# Antidepressant and Anxiolytic Effects of L-Methionine in the WAG/Rij Rat Model of Depression Comorbid with Absence Epilepsy

**DOI:** 10.3390/ijms241512425

**Published:** 2023-08-04

**Authors:** Karine Yu. Sarkisova, Alexandra V. Gabova, Ekaterina A. Fedosova, Alla B. Shatskova, Victor B. Narkevich, Vladimir S. Kudrin

**Affiliations:** 1Institute of Higher Nervous Activity and Neurophysiology of Russian Academy of Sciences, Butlerova Str. 5A, Moscow 117485, Russia; agabova@yandex.ru (A.V.G.); ekaterina5fedosova@rambler.ru (E.A.F.); all1133@mail.ru (A.B.S.); 2Federal State Budgetary Institution “Scientific Research Institute of Pharmacology named after V.V. Zakusov”, Baltiyskaya Str. 8, Moscow 125315, Russia; narvik@yandex.ru (V.B.N.); kudrinvs@mail.ru (V.S.K.)

**Keywords:** animal model, depression, anxiety, absence epilepsy, antidepressant, L-methionine, brain monoamine, WAG/Rij rat

## Abstract

Depression is a severe and widespread psychiatric disease that often accompanies epilepsy. Antidepressant treatment of depression comorbid with epilepsy is a major concern due to the risk of seizure aggravation. SAMe, a universal methyl donor for DNA methylation and the synthesis of brain monoamines, is known to have high antidepressant activity. This study aimed to find out whether L-methionine (L-MET), a precursor of SAMe, can have antidepressant and/or anxiolytic effects in the WAG/Rij rat model of depression comorbid with absence epilepsy. The results indicate that L-MET reduces the level of anxiety and depression in WAG/Rij rats and suppresses associated epileptic seizures, in contrast to conventional antidepressant imipramine, which aggravates absence seizures. The antidepressant effect of L-MET was comparable with that of the conventional antidepressants imipramine and fluoxetine. However, the antidepressant profile of L-MET was more similar to imipramine than to fluoxetine. Taken together, our findings suggest that L-MET could serve as a promising new antidepressant drug with anxiolytic properties for the treatment of depression comorbid with absence epilepsy. Increases in the level of monoamines and their metabolites—DA, DOPAC, HVA, NA, and MHPG—in several brain structures, is suggested to be a neurochemical mechanism of the beneficial phenotypic effect of L-MET.

## 1. Introduction

Depression is a severe chronic disease that affects more than 350 million people worldwide, making it one of the most common psychiatric disorders [1]. Depression is the most frequent comorbidity of different forms of epilepsy, including absence epilepsy [2,3,4,5,6]. The prevalence rate of neuropsychiatric comorbidities in epilepsy is 30–35% [5]. It is believed that comorbid depression may impact the quality of life more than epileptic seizures per se [7]. Given this, antidepressants are often used in the clinic to relieve depressive symptoms. However, the effectiveness and safety of antidepressant therapy remain problems that have not been completely solved. Around 30–60% of patients with depression are resistant to antidepressant therapy. Of particular importance are the effectiveness and safety of antidepressant therapy in the treatment of depressive disorders comorbid with epilepsy. Side effects of antidepressants, in particular the occurrence of epileptic seizures, are the main reason for not treating depression [8]. Therefore, there is an urgent need to develop new antidepressants with new mechanisms of action, which could benefit both comorbid pathologies—depression and the associated epilepsy.

The Wistar Albino Glaxo/Rijswijk (WAG/Rij) rat strain is a well-validated genetic model of childhood absence epilepsy with mild depression-like (dysthymia) comorbidity [9,10,11,12,13]. The model meets all three necessary validity criteria to be an animal model of depression: similarity with depressive pathology in symptoms (face validity), sensitivity to conventional antidepressants (predictive validity), and neurochemical alterations in the brain (construct validity) [12]. Thus, WAG/Rij rats exhibit depression-like behavior: decreased investigative activity in the open-field test, increased immobility in the forced swimming test and decreased sucrose consumption and preference (anhedonia). In addition, WAG/Rij rats display helplessness, submissiveness, and an inability to make choices and overcome obstacles, which are typical for depressed patients. WAG/Rij rats are also sensitive to chronic, but not acute, antidepressant treatments like depressed humans [12]. A functional deficiency of the brain monoamine systems, which is characteristic of depressive disorders, is believed to underlie depression-like behavioral symptoms [9,14]. The good predictive validity of the WAG/Rij rat model was also illustrated using a large number of antidepressant drugs [15].

Spike-wave discharges (SWDs), the main hallmark of absence epilepsy, are expressed age-dependently both in WAG/Rij rats [16] and GAERS (Genetic Absence Epilepsy Rats from Strasbourg), another genetic model of absence epilepsy [17]. There is evidence that a reduced dopamine (DA) tone of the brain mesolimbic system may contribute to depression-like behavioral comorbidity [18] and the genesis of SWDs [19], suggesting a close and causal relationship between these pathologies. Interestingly, no phenotypic expression of depression-like behavioral symptoms was found in pre-symptomatic (36-day-old) WAG/Rij rats. Depressive-like symptoms appear at the age of 3 months when SWDs start to be clearly expressed. Then, with age, the depressive symptoms increase as absence seizures are aggravated, and at the age of 6–7 months, the pathologic phenotype in WAG/Rij rats is fully expressed [13,18]. Interestingly, the aggravation of phenotypic manifestations of absence epilepsy and comorbid depression was accompanied by the deepening of DAergic insufficiency in the brain structures of WAG/Rij rats [18].

Although a pathologic phenotype in WAG/Rij rats is genetically determined, previous data have shown that the absence epilepsy and depression-like comorbidity can be altered by early postnatal environmental impacts (maternal care, maternal diet, neonatal maternal separation, and neonatal handling), indicating that epigenetic mechanisms might be involved [13,20].

Even though genetic factors play an important role in the pathogenesis of depression [21], epigenetic mechanisms make a significant contribution to the manifestation of depressive disorders [22,23,24]. Epigenetic modifications alter gene expression without affecting the DNA sequence. Epigenetic modifications can include DNA methylation, histone modifications, chromatin remodeling, non-coding RNA, and miRNA [1,25]. Impairments to DNA methylation were found in animals with depressive-like behavior and patients with depressive disorders [1,26]. In addition, it has been shown that epigenetic factors, in particular DNA methylation, play an important role in the mechanisms of antidepressant action [23,27,28]. Epigenetic mechanisms also underlie epileptogenesis and the effects of anti-seizure medications [29,30,31].

The potential role of the insufficiency of methyl-group donors in the pathogenesis of depression was confirmed by the effectiveness of L-methyl folate in the treatment of depressive disorders in humans [32]. It was found that the level of DNA methylation in the amygdala was reduced in a rat line genetically predisposed to increased anxiety, depression-like behavior, and low sociability (Low Responders). In the anxious and depressive line of rats, compared with “normal” rats, differences in the methylation of 793 DNA sites were found. Enhancing DNA methylation (via increased dietary methyl-group donors) improved the anxiety/depression-like phenotype. Conversely, dietary methyl donor depletion exacerbated depression-like behavior in the forced swimming test [26]. A change in the DNA methylation in the nucleus accumbens in mice led to antidepressant or pro-depressant behavioral effects in the forced swimming test, depending on the direction of the changes in the level of DNA methylation [33]. These data indicate the possible involvement of DNA methylation in the nucleus accumbens in the regulation of depression-like behavior in the forced swimming test.

S-adenosyl-L-methionine (SAMe) is a universal donor of methyl groups necessary for DNA methylation and the synthesis of the neurotransmitters DA, noradrenaline (NA), serotonin (5-HT), the deficiency of which in the brain leads to depressive disorders. A reduced SAMe content was found in depressed humans [34]. In placebo-controlled studies in patients with major depressive disorders, it has been demonstrated that SAMe has high antidepressant activity and no side effects, characteristic of treatment with traditional antidepressants. These data allowed the authors to suggest that increasing the SAMe level in the brain may be an effective new strategy for the treatment of depressive disorders that are primarily resistant to antidepressant therapy [34,35]. However, the idea of the potential antidepressant-like effect of methyl-group donors cannot be considered fully proven. Even though research in this area has been conducted for more than 20 years, there are many unresolved issues. One such question is the question of the antidepressant efficacy of methyl-group donors in the treatment of other forms of depressive disorders, including those associated with epilepsy. The precursor of SAMe is L-methionine (L-MET). The injection of L-MET more effectively increases the level of SAMe in the brain compared with the injection of SAMe [36]. In addition, L-MET enters the brain from the blood faster than SAMe [37] and, in contrast to SAMe, is more stable [36]. These data suggest that L-MET should also have antidepressant activity, similar to SAMe [38]. However, there is no experimental evidence for this assumption. Moreover, it is not clear whether L-MET can have antiepileptic activity. Several studies focus on the effect of L-MET on locomotor and exploratory activity, anxiety level, memory, and convulsive seizures (kainate model of temporal lobe epilepsy) [39,40]. However, there are no studies of the effects of L-MET on depression associated with absence epilepsy in experimental models of this pathology.

This study aimed to test the assumption that L-MET can have an antidepressant effect in the WAG/Rij rat model of depression comorbid with absence epilepsy.

To achieve the aim, several questions were asked:Does L-MET exhibit an antidepressant and/or anxiolytic activity in tests relevant to the assessment of anxiety and depression?Is the antidepressant effect of L-MET comparable with that of conventional antidepressant drugs, such as the tricyclic antidepressant imipramine and the selective serotonin reuptake inhibitor fluoxetine?What is the effect of L-MET on the associated absence epilepsy, and is it different from the effect of the reference antidepressant imipramine?Are the effects of L-MET on depression-like behavior and absence seizures related to alterations in the brain monoamine systems?

## 2. Results

### 2.1. The Effect of L-MET on the Anxiety Level in WAG/Rij Rats

In the open-field test, the one-way ANOVA showed a significant effect of L-MET on the latency to leave the center (F(1,14) = 5.83, *p* < 0.05), the number of the squares crossed (F(1,14) = 7.46, *p* < 0.05), and the number of center entries (F(1,14) = 16.19, *p* < 0.001). In L-MET-treated WAG/Rij rats, the latency to leave the center was lower, but the number of crossed squares and center entries was greater compared with their vehicle-treated counterparts. L-MET caused a tendency toward a smaller number of boli (Table 1).

In the light–dark test, the one-way ANOVA showed a significant effect of L-MET on the time spent in the light (F(1,14) = 4.32, *p* = 0.05) and the number of risk assessments (F(1,14) = 6.13, *p* < 0.05). In L-MET-treated WAG/Rij rats, the time in the light was greater than and the number of risk assessments was less than the vehicle-treated controls. The number of transitions between compartments tended to be lower (Table 1).

In the elevated plus-maze test, a one-way ANOVA showed a significant effect of L-MET on the time in open arms (F(1,14) = 4.81, *p* < 0.05), the number of hangings from the open arms (F(1,14) = 7.64, *p* < 0.05), and the number of boli (F(1,14) = 5.51, *p* < 0.05). In L-MET-treated WAG/Rij rats, the time in open arms and the number of hangings from the open arms were greater, but the number of boli was lower, compared with the vehicle-treated group. The number of risk assessments and rearings in the closed arms tended to be less than that of the vehicle-treated control group (Table 1).

### 2.2. The Effect of L-MET on Depression-like Behavior in WAG/Rij Rats in Comparison with the Effects of the Antidepressants Imipramine and Fluoxetine

In the forced swimming test, a one-way ANOVA showed a significant effect of L-MET on immobility time (F(1,13) = 4.63, *p* < 0.05), the duration of the first episode of active swimming (F(1,13) = 6.51, *p* < 0.05), and the number of dives (F(1,13) = 15.70, *p* < 0.01). L-MET did not significantly affect the duration of swimming (F(1,13) = 1.76, *p* = 0.21). L-MET treatment reduced the immobility time, increased the duration of the first episode of active swimming (“climbing”), and did not significantly affect the duration of swimming, indicating an antidepressant-like effect (Table 2). The conventional antidepressants imipramine and fluoxetine increased climbing and swimming, respectively, and both drugs reduced the immobility time compared with vehicle. Of note, the antidepressant profile of L-MET (decreased immobility time) was comparable to that of classical antidepressants but had a greater similarity with imipramine (increased duration of climbing and number of dives) than with fluoxetine (no substantial increase in the duration of swimming).

### 2.3. The Effect of L-MET on Absence Seizures in WAG/Rij Rats in Comparison with the Effect of the Antidepressant Imipramine

EEG studies have shown that chronic injection of L-MET at a dose of 50 mg/kg for 14 days reduces the number of SWDs in WAG/Rij rats. However, the effect of L-MET on the number of SWDs depended on the baseline severity of the absence seizures (the initial number of SWDs). L-MET decreased the number of seizures in a subgroup of WAG/Rij rats with an initially smaller number of seizures (59.33 ± 9.88), called “sensitive”, but did not change the number of discharges in a subgroup of WAG/Rij rats with an initially larger number of seizures (99.5 ± 15.54), called “insensitive”. Two-way ANOVA with repeated measures showed a significant effect of the initial number of seizures on the effect of L-MET on the number of SWDs (F(1,30) = 21.24, *p* < 0.001). A significant decrease in the number of SWDs was observed only in the “sensitive” subgroup of WAG/Rij rats and only on the 14th day of L-MET administration (Figure 1A). L-MET did not significantly affect the number of SWDs in the “insensitive” subgroup of WAG/Rij rats (Figure 1C) or on the mean duration of SWDs in both subgroups of WAG/Rij rats with initially shorter (Figure 1B) and longer (Figure 1D) mean duration of discharges. On the 7th day after cancellation of the L-MET administration, the number of SWDs in the “sensitive” subgroup of WAG/Rij rats did not significantly differ from the baseline level.

On the 14th day of L-MET treatment, a decrease in the number of well-developed mature SWDs registered before treatment (Figure 2A) was accompanied by the appearance of immature discharges (Figure 2B). Immature discharges have altered time-frequency dynamics (subpanel 2 of Figure 2B), frequency spectra (Figure 2B, subpanel 3), and morphological characteristics (Figure 2B, subpanel 4), indicating less severe absence seizures. Immature discharges were characterized by fewer sharp spikes in the spike-wave complex as well as the irregular presence of a wave (Figure 2B, subpanel 4).

Analysis of SWDs with a Fast Fourier Transform procedure revealed that on the 14th day of L-MET treatment, the averaged spectral power at the fundamental frequency and the first and second harmonics was significantly lower compared with the baseline (before treatment) and the 7th day of treatment. Even though on the 7th day after the cancellation of L-MET, the number of epileptic discharges was restored to the initial level, the averaged spectral power of discharges remained lower than the initial one and did not significantly differ from that on the 14th day of treatment (Figure 3).

Treatment with L-MET did not change the background EEG (Figure 4), indicating a selective effect on epileptic discharges.

The effect of imipramine on the number of SWDs in WAG/Rij rats depended on the treatment option (acute or chronic). The two-way ANOVA revealed a significant effect of the factors “drug” (F(1,20) = 5.22, *p* < 0.05) and “treatment option” (F(1,20) = 18.3, *p* < 0.001) as well as the interaction of these factors (F(1,20) = 16.66, *p* < 0.001) on the mean duration of the SWDs. Acute administration of imipramine reduced the mean duration of absence seizures compared with acute and chronic vehicle administration, while chronic treatment did not significantly affect this measure (Figure 5B). The two-way ANOVA showed a non-significant effect of the factors “drug” (F(1,20) = 1.67, *p* = 0.21) and “treatment option” (F(1,20) = 3.8, *p* = 0.06) as well as the interaction of these factors (F(1,20) = 3.66, *p* = 0.07) on the number of seizures. The effect of acute and chronic imipramine administration on the number of SWDs was the opposite: acute treatment reduced the number of seizures, while chronic administration increased the measure of absence seizure severity (Figure 5A).

The one-way ANOVA showed a significant effect of drugs on the amplitude (F(2,66) = 33.86, *p* < 0.001) and asymmetry index (F(2,66) = 25.19, *p* < 0.001) of epileptic discharges. L-MET treatment reduced the amplitude and asymmetry index of SWDs compared with vehicle-treated and imipramine-treated WAG/Rij rats. Although imipramine treatment also reduced the amplitude of epileptic discharges, the effect of L-MET was greater than that of imipramine. Unlike L-MET, imipramine did not significantly change the asymmetry index of the SWDs (Table 3).

The increase in the number of SWDs caused by 14-day treatment with imipramine indicates its negative effect on absence seizures. Even though the mean duration of SWDs after chronic imipramine treatment did not significantly change compared with vehicle administration (Figure 5B), the interval between discharges decreased, which led to the fusion of individual discharges in one long discharge (Figure 6).

The duration of individual discharges after imipramine administration could reach 100 s or even more, while the mean duration of the discharges after vehicle administration did not exceed 20 s. Thus, the negative effect of imipramine treatment was also manifested in the form of a fusion of individual discharges into one very long discharge, resembling absence status epilepticus. To determine whether imipramine can have a pro-epileptic effect on absence seizures, the effect of a two-fold higher dose of imipramine (30 mg/kg) was studied. On the 4th day of imipramine injection at a higher dose, atypical (unusual for absence epilepsy) epileptic discharges appeared on the EEG, accompanied by convulsive twitching of the head (Figure 7).

Atypical (abnormal) epileptic discharges had an amplitude (about 3000 µV) several times higher than the amplitude of typical SWDs (usually no more than 1000 µV), as well as poorly expressed asymmetry. In addition, they were characterized by the disorganization of their time–frequency dynamics and a substantial change in the power spectra, as evidenced by the wavelet (Figure 7B) and Fourier analysis (Figure 7C). Abnormal morphological characteristics of atypical discharges (subpanels 1, 2, 3 of Figure 7C) indicate disorganized spike-and-wave activity. In an atypical discharge, three consecutive parts can be distinguished, which differ in morphology. The initial part of the discharge (Figure 7D, subpanel 1) consisted mainly of spikes, and waves were absent. The middle part of the discharge consisted of spikes and waves of inverted polarity with a frequency of up to 2–3 Hz (instead of the 7–8 Hz that is characteristic of a typical SWD). The abnormal discharge ended with the alterations of two spikes and several waves with a frequency of less than 1 Hz. The spikes were accompanied by brief, involuntary, head twitching (focal myoclonus), reminiscent of the imipramine-induced myoclonus described earlier [41]. Taken together, these data suggest that imipramine at a certain dose can aggravate absence seizures and may have a pro-convulsive effect in the WAG/Rij rat model of absence epilepsy.

To determine whether L-MET can have a pro-epileptic effect like imipramine, the effect of two-fold, four-fold, and six-fold higher doses of the drug (100, 200, and 300 mg/kg) on the number of SWDs was compared with the effect of L-MET at a dose of 50 mg/kg. Saline served as the control drug. Different doses of L-MET were tested in the same animals. The different doses of drugs were administered at intervals of at least 7 days [41]. The results showed that, unlike imipramine, all doses of L-MET did not increase the number of SWDs. L-MET at doses of 100 and 300 mg/kg did not change the number of SWDS compared to the initial level taken as 100%. L-MET at doses of 50 and 200 mg/kg decreased the number of SWDs by the 2nd and 4th hour after administration, respectively. The dose of 50 mg/kg proved to be the most effective because it caused a decrease in the number of absence seizures by more than two-fold compared with the baseline (Figure 8).

### 2.4. The Effect of L-MET on Monoamines and Their Metabolite Content in Brain Structures

Biochemical data have shown that chronic administration of L-MET causes an increase in the level of monoamines and their metabolites in all studied brain structures in WAG/Rij rats (Table 4). Of note, the content of DA and its metabolites were more affected than the content of NA and 5-HT.

In the prefrontal cortex, chronic injections of L-MET induced an increase in the content of NA (F(1,14) = 8.49, *p* < 0.05). Changes in other monoamines and their metabolite levels were insignificant.

In the nucleus accumbens, L-MET treatment significantly increased the content of the DA metabolite DOPAC (F(1,14) = 4.8, *p* < 0.05) and, at the trend level, the content of DA (p(U) = 0.09). L-MET also caused a tendency to increase the level of the NA metabolite MHPG (F(1,14) = 3.02, *p* = 0.1) and the ratio of 5-HIAA to 5-HT (F(1,14) = 3.0, *p* = 0.1; p(U) = 0.028). 5-HIAA is the primary product of the enzymatic degradation of serotonin by monoamine oxidase A (MAO-A). The 5-HIAA-to-5-HT ratio is an indication of the serotonin turnover rate in the brain and serves as an estimated index for assessing the rate of release of 5-HT into the synapse.

In the striatum, L-MET treatment significantly elevated the level of HVA, the end product of DA degradation (F(1,14) = 22.15, *p* < 0.001), the HVA/DA index (F(1,14) = 4.61, *p* < 0.05) and, at a tendency level, the content of DA (F(1,14) = 4.0, *p* = 0.06) and its catabolic product DOPAC (F(1,14) = 4.10, *p* = 0.06).

In the hypothalamus, L-MET significantly increased the content of NA (F(1,14) = 14.18, *p* < 0.01); the metabolic product of NA, MHPG, (F(1,14) = 28.36, *p* < 0.001); and, at the level of a tendency, the content of DA (F(1,14) = 4.07 *p* = 0.06) and the 5-HIAA/5-HT index (F(1,14) = 3.00, *p* = 0.1).

In the hippocampus, L-MET caused a tendency to increase the levels of DA (F(1,14) = 3.00, *p* = 0.1), DOPAC (F(1,14) = 3.37 *p* = 0.08), and 3-methoxytyramine (MT) (F(1,14) = 3.00 *p* = 0.1).

## 3. Discussion

The present study describes for the first time the anxiolytic and antidepressant effects of L-MET treatment in the WAG/Rij rat model of depression comorbid with absence epilepsy. The results indicate that L-MET not only reduces the level of anxiety and depression in WAG/Rij rats but also suppresses associated epileptic seizures, in contrast to conventional antidepressants, especially tricyclic and tetracyclic, many of which can decrease the seizure threshold and aggravate epileptic seizures [7,41,42,43,44,45].

The anxiolytic effect of L-MET is evidenced by a reduction in anxiety measures in three well-validated tests for the assessment of anxiety-related features of behavior: the open-field, light–dark choice, and elevated plus-maze tests [46].

In the open-field test, L-MET caused a significant increase in the number of crossed squares, center entries, and unassisted (unsupported) rearing (the measures of exploratory motivation) and a tendency to decrease the number of fecal boli (a measure of fear/anxiety provoked by a new unfamiliar open space). Interestingly, L-MET treatment did not affect assisted (supported) rearing. The open-field test is a conflict test based on opposing motivations: to explore a new environment (exploratory motivation) and to avoid a central (more “dangerous”) area due to fear/anxiety. A classic exploratory behavior recorded in the open field is rearing behavior. However, previous studies have shown that supported rearing (where the animal rears against the wall of the arena) and unsupported rearing (where the animal rears without contacting the wall of the arena) are two different behavioral traits in the open-field test [47,48]. Supported rearing is more associated with “activity”, while unsupported rearing is highly sensitive to contextual parameters that impact the aversiveness of the test (e.g., light intensity), similar to “time in the center”, a parameter typically associated with emotionality and anxiety [48]. Moreover, unsupported rearing negatively correlates with classic measures of anxiety in the open-field test, such as defecation [47]. Therefore, an increased number of center entries, unassisted rearing, and decreased fecal boli in the open-field test indicate a decrease in anxiety induced by chronic treatment with L-MET in WAG/Rij rats.

In the light–dark choice test, the anxiolytic effect of L-MET is evidenced by an increase in the time spent in the lit compartment and a decrease in the number of risk assessments, an ethologically relevant measure of anxiety [49,50]. In L-MET-treated WAG/Rij rats, the number of transitions between light and dark compartments tended to be greater compared with a vehicle-treated control group.

In the elevated plus-maze, L-MET induced an increase in the time spent in open arms, the number of head dips from the open arms, and a decrease in the number of boli. Observing stress-related anxiety in rodents typically relies on species-specific (ethologically driven) behaviors such as increased risk assessment, the reduction of exploration, seeking shelter, escape, and defecation [51]. Therefore, an increase in the duration of open arms and the number of head dips are both behaviors that point to decreased stress-related anxiety. A reduced number of fecal boli also indicates a reduced level of anxiety. Of note, L-MET did not significantly change the number of transitions between arms (a measure of locomotor activity in this test), thereby excluding the possibility that the anxiolytic-like effect of L-MET in the elevated plus-maze may be a consequence of changes in general motor activity.

In the forced swimming test, L-MET treatment reduced the immobility time, increased the duration of the first episode of active swimming (climbing), and did not significantly affect the duration of swimming compared with saline treatment, indicating an antidepressant-like effect. Of note, the antidepressant profile of L-MET (decreases in immobility) was comparable with that of classical antidepressants such as imipramine and fluoxetine. However, the antidepressant-like effect of L-MET in the forced swimming test was more similar to the effect of the tricyclic antidepressant imipramine than that of the selective serotonin reuptake inhibitor fluoxetine. Thus, L-MET, like imipramine, increased the duration of climbing and did not affect the duration of swimming, while fluoxetine increased the duration of swimming and did not affect the duration of climbing. It should be noted that the effect of L-MET on anxiety levels in WAG/Rij rats also has a greater similarity with the effect of imipramine than of fluoxetine. L-MET, like imipramine [52], showed an anxiolytic effect, but fluoxetine did not [53]. This probably means that L-MET, like imipramine, is an antidepressant drug with anxiolytic action.

One of the most important characteristics of the beneficial action of L-MET in the WAG/Rij rat model of depression and absence epilepsy comorbidity is the combination of anxiolytic and antidepressant effects with an overwhelming effect on absence seizures. This seems to indicate that L-MET affects the common mechanisms underlying absence epilepsy and its neuropsychiatric comorbidities. Previous behavioral, electrophysiological, and pharmacological studies have demonstrated functional deficits in the brain DAergic system responsible for depression-like behavior in WAG/Rij rats [12,14,18,52]. A reduced DAergic tone of the mesolimbic brain system was shown to be associated with both absence seizures [19] and depression-like comorbidity [14,18] in WAG/Rij rats. Systemic administration of the mixed dopamine D1/D2 receptor agonists resulted in a reduction of SWDs [54] and depression-like comorbidity [52], while DA antagonists increased the number of SWDs [54] and depression-like behavioral symptoms [52] in WAG/Rij rats. The basal ganglia are known to modulate absence seizures [55,56]. At the same time, DAergic neurotransmission within the dorsal and ventral striatum plays an important role. The ventral striatum (nucleus accumbens) is critical in the control of absence seizures [55] and depression-like behavioral symptoms in WAG/Rij rats [14]. A decrease in DAergic activity in the striatum is believed to predispose to cortical hyper-excitability and epilepsy [57]. Previous biochemical studies have shown reduced DA content in the nucleus accumbens and the striatum of WAG/Rij rats [14,18]. Therefore, an increase in the levels of DA and its metabolites (DOPAC and HVA) in the nucleus accumbens and striatum caused by L-MET may explain the correction of symptoms of absence epilepsy and comorbid depression in WAG/Rij rats.

L-MET also increased the NA content in the prefrontal cortex and the content of NA and its metabolite, MHPG, in the hypothalamus. There is convincing evidence that the brain’s NAergic system is the intrinsic anti-epileptic mechanism. NA regulated the excitability of cortical neurons [58], adjusting the balance between excitation and inhibition, the impairment of which plays an important role in the pathogenesis of absence epilepsy. Accumulating evidence suggests that stimulation of NAergic signaling inhibits seizures, whereas depletion of NA increases seizure susceptibility and accelerates epileptogenesis in many animal models. In genetically engineered mice that lack NA, an increased seizure susceptibility to different convulsive stimuli has been demonstrated [59]. Inhibition of NA by clonidine aggravated absence seizures in WAG/Rij rats [60]. Interestingly, increased levels of NA in the prefrontal cortex and hypothalamus were detected in WAG/Rij rats at an age when SWDs had not yet appeared. With age, the absence seizures appeared, accompanied by a decrease in the content of NA in the prefrontal cortex and hypothalamus [18]. These data allow us to assume that an increase in the level of NA in the prefrontal cortex caused by L-MET reduces the excitability of cortical neurons leading to the suppression of SWDs. NA is believed to play an important role in the pathogenesis of depression and the mechanisms of action of antidepressant drugs [61]. Moreover, NA modulated neuronal activity and is essential for the “optimal” functioning of the prefrontal cortex involved in the regulation of the arousal level, decision making, working memory, cognition, and attention. When the “optimal” level of NA release in the prefrontal cortex is disrupted, impairments of the top-down control over other brain regions occur, leading to behavioral disorders; in particular, increased anxiety [62]. The hypothalamus and hippocampus are related to some aspects of depression [63]. Therefore, it can be assumed that increases in the content of NA and DA in the hypothalamus and of DA, DOPAC, and 3-MT in the hippocampus may underlie the antidepressant-like effect induced by L-MET in WAG/Rij rats. The lack of necessary data does not allow us to link the increase in 5-HT metabolism (5-HIAA/5-HT) in the nucleus accumbens and hypothalamus with the positive effect of L-MET on depression and its associated absence seizures in WAG/Rij rats.

L-MET is the immediate precursor of SAMe and the end product of the one-carbon cycle [64]. SAMe is a major donor of the methyl groups required for DNA methylation, which is one of the most studied epigenetic mechanisms that alter gene expression without changing the DNA sequence. Moreover, SAMe affects not only DNA methylation leading to epigenetic modifications of gene expression but is also involved in the synthesis of many neurotransmitters in the brain (DA, NA, 5-HT) [65], the insufficiency of which plays an important role in the pathogenesis of depression [66]. The monoamines DA and NA are synthesized from the amino acid tyrosine in a series of chemical reactions dependent on tyrosine hydroxylase. 5-HT is synthesized from the amino acid tryptophan, and the rate-limiting step is catalyzed by tryptophan hydroxylase. SAMe functions as a methyl-donating cofactor in the rate-limiting step of the synthesis of the monoamines DA and 5-HT [64]. Enhancement of SAMe levels permits it to act as a cofactor of COMT, decreasing COMT enzyme activity and thereby the degradation of catecholamines [67]. Therefore, SAMe can be regarded as a treatment option for depressive disorders that increase monoamines since low levels of SAMe, elevated homocysteine, and low 5-HT, DA, and NA are usually found in depressive patients [68]. Indeed, SAMe is an effective antidepressant drug [64,69] as it is well tolerated and may have a relatively faster onset of action than conventional antidepressants [64]. In addition, an antiepileptic and memory-enhancing effect of SAMe administration in a PTZ-induced kindling model of epilepsy has also been demonstrated [70]. In our studies, we used L-MET at a dose (50 mg/kg) that increases the brain level of SAMe. Interestingly, L-MET at a dose of 50 mg/kg caused a greater increase in the content of SAMe in the brain than at a dose of 100 mg/kg. Moreover, L-MET was more efficient than SAMe in increasing the level of SAMe in the brain [36]. This means that the antidepressant and anxiolytic effects of L-MET treatment found in our study were most likely caused by an increase in the SAMe content in the brain of WAG/Rij rats. The increased level of monoamines and their metabolites in the brain of WAG/Rij rats after L-MET treatment supports this assumption. The fact that L-MET at the dose of 50 mg/kg was the most effective in suppressing SWDs in WAG/Rij rats and that this effect appeared by the 4th hour after administration confirms previously published data [36]. Our findings concerning the anxiolytic-like effect of L-MET in WAG/Rij rats are consistent with the data of others who have shown an anxiolytic-like effect of low doses (5 and 10 mg/kg) of L-MET treatment comparable to that of diazepam in the elevated plus-maze test in Wistar rats [39]. It should be noted that the anxiolytic and antidepressant-like effects of SAMe have been well-documented previously [39,64,66,69]. However, to our knowledge, this is the first study that has shown the anxiolytic and antidepressant effects of L-MET in the WAG/Rij rat model of depression comorbid with absence epilepsy. Moreover, the favorable effects of L-MET on psychiatric comorbidities were accompanied by favorable effects on associated absence seizures, in contrast to the effects of many conventional antidepressant drugs [42,44]. Under certain conditions (dose, treatment option, duration of treatment, etc.), the tricyclic antidepressant imipramine, according to the data of the present study, as well as the selective serotonin reuptake inhibitor fluoxetine, according to data from other authors [15], may aggravate absence seizures in WAG/Rij rats. The results of the present study are in line with the literature data showing that older antidepressants, in particular tricyclic ones, can increase the occurrence of epileptic seizures [71].

One important question is why the smallest dose of L-MET was the most effective in suppressing absence seizures. First, apparently, because it is this dose of L-MET that increases the SAMe content in the brain but not in the blood. These data indicate that L-MET causes an increase in the local synthesis of SAMe in the brain, rather than an increase in the synthesis of SAMe peripherally, with subsequent transport of SAMe from the periphery to the brain [36]. We agree with the authors’ assumption that a smaller dose of L-MET increases the level of SAMe in the brain because methionine adenosyltransferase, the enzyme that produces SAMe from methionine, is not normally saturated with methionine [36]. Higher doses of L-MET (above the optimal dose) may cause a decrease in the synthesis of SAMe due to substrate inhibition. It is known that many enzymes are inhibited by their substrates, leading to velocity curves that rise to a maximum and then descend as the substrate concentration increases. Substrate inhibition often has important biological functions. For example, substrate inhibition of tyrosine hydroxylase results in a steady synthesis of DA despite large fluctuations in tyrosine due to meals. Nonetheless, this does not exclude the idea that substrate inhibition of methionine adenosyltransferase also leads to a steady synthesis of SAMe in the brain, regardless of fluctuations in the level of methionine.

In the present study, it has been also demonstrated that one of the mechanisms of the beneficial phenotypic effects of L-MET is an increase in the level of monoamines and their metabolites, mainly DA, DOPAC, HVA, NA, and MHPG, in several brain structures. Reduced monoaminergic tone, especially mesolimbic DAergic insufficiency, has been earlier reported to be associated with depression-like pathology in WAG/Rij rats [14,18].

In our previous work, a maternal diet enriched with methyl-group donors and cofactors of the one-carbon cycle (choline, betaine, folic acid, vitamin B12, L-MET, zinc) during the perinatal period enhanced the DAergic tone of the brain mesolimbic system [72], increased the expression of selected genes potentially representing the cause of the pathology, and suppressed the occurrence of absence seizures and comorbid depression in adult offspring of the WAG/Rij rats [20]. SAMe treatment of pregnant submissive mice, as a model of depression, alleviated depressive-like behavioral symptoms and changed the level of brain monoamines and the expression of genes related to monoamine metabolism in adult offspring [73]. These data indicate that treatment with methyl-group donors during neurodevelopment, which is the most sensitive period for epigenetic modifications, can exert long-lasting disease-modifying effects and, therefore, may represent a promising new approach to the treatment of depressive disorders related or unrelated to seizures. Here we have shown that short-term treatment of adult WAG/Rij rats with a methyl-group donor can be as effective in alleviating depression and associated absence seizures as early and long-term treatment [20]. The question of whether L-MET treatment causes a change in the expression of genes, including those related to the metabolism of monoamines in the brain, as well as what their epigenetic mechanisms are, awaits further research.

## 4. Materials and Methods

### 4.1. Animals

Males of inbred WAG/Rij rats were used as experimental subjects. WAG/Rij rats were born and raised at the Institute of Higher Nervous Activity and Neurophysiology of the Russian Academy of Sciences and represented approximately the 72nd generation from parents originally obtained from Radboud University Nijmegen (Nijmegen, The Netherlands). All rats were kept under a 12/12 h light–dark cycle (lights on at 8.00 a.m.). Animals were housed in standard plastic cages in groups of 3–4 animals per cage. Food and tap water were available ad libitum. Experiments were performed according to the European Union Directive 2010/63/EU on the protection of animals used for scientific purposes. Animal care and use were in accordance with the institutional policies and guidelines. The experimental protocol was approved by the Ethical Committee of the IHNA (protocol № 5 of 2 December 2020). All efforts were made to minimize the number of animals used in experiments and their suffering from experimental procedures.

### 4.2. Drug Administration

L-MET (Merck KGaA, Darmstadt, Germany) was injected intraperitoneally (i.p.) at a dose of 50 mg/kg for 14 days. The 50 mg/kg dose was chosen because it has been shown that it is at this dose that L-MET causes an increase in the SAMe level in the brain. Other doses (25 mg/kg and 100 mg/kg) were less efficient [36]. In addition to the chronic administration of L-MET at a dose of 50 mg/kg, a single injection was used at doses of 50, 100, 200, and 300 mg/kg. The tricyclic antidepressant imipramine hydrochloride (Sigma-Aldrich, St. Louis, MO, USA) and the selective serotonin reuptake inhibitor fluoxetine (Sigma-Aldrich, St. Louis, MO, USA) were given i.p. at a dose of 15 mg/kg for 14 days. Both drugs in the selected dose caused a distinct antidepressant-like effect [9,53] in the WAG/Rij rat model. For the acute imipramine treatment, rats were first given daily i.p. injections of saline for 11 days and, on the next 3 days (days 12–14), i.p. injections of imipramine (15 mg/kg/day). Drugs were diluted in saline. All administrations were in a volume of 1 mL/kg body weight. Control animals were treated i.p. by saline (vehicle) using the same volume as the drugs.

### 4.3. Behavioral Testing, EEG Registration, and Brain Monoamine Level Measurements

Adult 7-month-old WAG/Rij rats (vehicle-treated and L-MET-treated) were randomly divided into 2 groups: one of them was subjected to EEG registration and behavioral testing (n = 48) and the other (n = 16) to measuring the level of brain monoamines and their metabolites. Animals intended for biochemical studies were not subjected to behavioral testing to prevent any possible effects of testing on the brain monoamines and their metabolite levels.

#### 4.3.1. EEG Registration and Analysis

Stereotactic surgery was performed under chloral hydrate anesthesia (400 mg/kg, i.p.). Rats were equipped with bilateral epidural electrodes (stainless steel screws) over the frontal (AP 2 mm, L 2.5 mm) and occipital (AP −6 mm, L 4 mm) cortex. The electrodes were implanted into small round holes (0.8 mm in diameter) in the skull and fixed with dental acrylic. EEG registration for 3 h per day (from 16.00–19.00) was conducted in freely moving animals using the wireless 8-channel biopotential measurement system “BR8V1” based on Texas Instruments ADS1298 Analog Front-End. The EEG was recorded monopolarly, and a reference electrode was placed over the cerebellum. Animals were placed in Plexiglas recording cages (15 cm × 30 cm × 26 cm) and habituated to the experimental situation for 1 h before the beginning of the recording session. SWDs were detected in the EEG as repetitive trains of sharp asymmetric large-amplitude spikes and slow waves lasting ≥1 s with amplitude at least two-fold higher than the baseline EEG signal. Calculation of SWDs was performed by a “blinded” unbiased expert. The severity of absence epilepsy was assessed by the number and mean duration of the SWDs. The mean duration of SWDs was calculated as the ratio of the total duration to the number of SWDs. The number and mean duration of the SWDs were calculated at baseline, on the 7th and 14th days of L-MET injections, and on the 7th day after L-MET cancellation. The time–frequency dynamics of the SWDs were analyzed using the complex wavelet Morlet. The frequency spectra of the EEG with SWDs were computed using the Welch method (Fast Fourier Transform procedure based on Hanning analysis with a fixed time-window length set at 2 s; the overlapping of the window was 7/8), and then the spectra were averaged.

Of note, the 2 s time window supplies a frequency resolution of 0.5 Hz. The SWDs’ time–frequency dynamics and spectral power were compared between 2 groups of WAG/Rij rats (drug-treated and vehicle-treated). The averaged power spectra of the background EEG during quiet wakefulness (fragments lasting 8–10 s and separated from SWDs at least by 2 s) were compared between the experimental and control groups of animals. In addition to the spectral power density, the SWDs’ amplitude (μV) and asymmetry index (the ratio of the negative component to the sum of negative and positive components of the SWDs in %) were calculated. Custom-made software was used for the semiautomatic measurements of the SWD amplitude and asymmetry index. The amplitudes of all spikes in a whole SWD were computed and then averaged.

#### 4.3.2. Behavioral Testing, Anxiety Level, and Depression-Like Symptoms

L-MET-treated and vehicle-treated WAG/Rij rats were compared for differences in behavioral tests relevant to anxiety and depression. The level of anxiety was determined in the light–dark choice, open-field, and elevated plus-maze tests. The forced swimming test assessed depression-like behavior in WAG/Rij rats. Behavioral tests were carried out in compliance with the recommendations developed for testing epileptic animals [46].

##### Light–Dark Choice Test

The apparatus consisted of two compartments with openings between them. The large (36 cm × 18 cm) compartment was light (100 lx) and the small (18 cm × 17 cm) one was dark (<5 lx). Each rat was placed in the light compartment facing away from the opening, and the following behavioral reactions were measured for 5 min: latency of entering the dark compartment, the time spent in each compartment, the number of transitions between compartments, and the number of risk assessments (aborted attempts to enter into the light compartment). The shorter the time spent in the light compartment, the lower the number of transitions between compartments, and the greater the number of risk assessments (an ethologically relevant measure of anxiety), the higher the level of anxiety in the test and vice versa [9,46].

##### Open-Field Test

The apparatus was a circular arena, 96 cm in diameter with a 40 cm wall, divided into 16 sectors and 3 circles. The central circle with a diameter of 32 cm was considered the center of the field. Animals were placed in the center of the field, and the following variables were recorded for 5 min: latency to leave the center, the number of crossed squares, the number of rearings (assisted and unassisted), center square entries, grooming reactions, and fecal boli (defecation). The open field was cleaned after testing each rat with 10% ethanol and wiped thoroughly to remove the odor. The test room had a dim illumination (40 W) for decreasing the aversiveness of the test. The lower exploratory activity in the open field, especially in the central area, is commonly ascribed to a higher level of anxiety and vice versa [9,46].

##### Elevated Plus-Maze Test

The elevated plus-maze is a standard and widely used test to assess anxiety-like behavior in rodents. Anxiety-like behavior is characterized by an increased avoidance of open arms compared with closed arms. The elevated plus-maze used in our study was constructed of dark gray polyvinyl chloride (OpenScience Ltd., Moscow, Russia). It consisted of two opposing open arms without side walls (length 50 cm, width 10 cm), two opposing enclosed arms (length 50 cm, width 10 cm) with side walls (height 38 cm), and a central square (10 cm × 10 cm). Two lights illuminated the maze (80 lx over the open arms, 5 lx over the closed arms). The maze was elevated 73 cm above the floor. The test duration was 3 min. Rats were brought to the test room 15 min before the start of the test (habituation period). Each rat was placed in the center of an elevated plus-maze facing a closed arm, and the following behavioral measures were recorded: the number of entries into and time (s) spent on each arm and central square, the number of head dips from the open arms, the number of rearings in the open and closed arms, the number of risk assessments (aborted attempts to enter into open arms), the number of transitions between open and closed arms, the number of grooming episodes, and the number of fecal boli. The maze was cleaned with 10% ethanol between trials to minimize scent trails. The shorter the time spent in open arms and the central square, the greater the number of risk assessments, and the higher anxiety and vice versa [9,50,74,75]. The total number of entries into open and closed arms (transitions between arms) is usually used as an index of locomotor activity in the test [74].

##### Forced Swimming Test

The forced swimming test is a standard test to measure depression-like behavior in experimental animals. This test is widely used to assess the antidepressant potential of various pharmacological drugs. In this study, the forced swimming test for the assessments of stress coping, “behavioral despair” or depression-like behavior was modified from the test originally described by Porsolt [76]. Only a single test session without a pre-test was used [9,10,20,72]. The apparatus was a cylinder (height 47 cm, inside diameter 38 cm) containing 38 cm of tap water maintained at 24 ± 1 °C. The behavior of rats in the forced swimming test was recorded by a video camera. Rats were individually forced to swim, and the following behavioral measures were recorded for 5 min: the duration of passive swimming (immobility), the duration of the first episode of active swimming (climbing), the duration of swimming, and the number of dives and boli. Immobility was defined as no movements by the rat (floating vertically in the water, front limbs are immovable and clasped to the breast, and nose is kept above the water surface). Climbing (“struggling”, jumping) was defined as upward-directed strong movements of the front limbs breaking the surface of the water that resemble scratching the wall of the container. All other less vigorous movements on the water surface throughout the water tank, with the rat in a horizontal position, were defined as swimming [9,10,20,72]. Increased immobility and decreased active behaviors (climbing, swimming, diving) in this test are considered to be indicative of a depression-like phenotype. The forced swimming test (Porsolt test of “behavioral despair”) is stressful for animals, so it was the last test performed to prevent any effect on the level of anxiety. The effect of L-MET was compared with that of the conventional antidepressants imipramine and fluoxetine.

#### 4.3.3. Brain Monoamines and Their Metabolite Level Measurements

Vehicle-treated and L-MET-treated WAG/Rij rats were decapitated with a guillotine for laboratory rodents. Then, the brain structures (prefrontal cortex, hippocampus, striatum, nucleus accumbens, and hypothalamus) were removed on ice, frozen in liquid nitrogen, and weighed. The samples were stored in liquid nitrogen and later assayed for levels of monoamines and their metabolites using high-pressure liquid chromatography (HPLC) with electrochemical detection. The details of the method used have been described earlier [14,18]. Briefly, the brain tissue was homogenized (Potter homogenizer, glass–Teflon) in 1.0 mL 0.1 M HCIO_4_ containing 3,4-dihydroxy benzylamine (0.5 nmol/mL) as an internal standard and centrifuged at 10,000× *g* for 10 min at 4 °C. The supernatant (20 µL) was removed and analyzed by HPLC to determine the concentration of noradrenaline (NA), 4-hydroxy-3-methoxyphenylglycol (MHPG), dopamine (DA), 3,4-dihydroxyphenylacetic acid (DOPAC), homovanillic acid (HVA), 5-hydroxytryptamine or serotonin (5-HT), 5-hydroxyindolacetic acid (5-HIAA). Chromatograph LC-304 T (BAS, West Lafayette, IN, USA) with analytical column ReproSil-Pur (ODS-3, 100 × 4 mm, 3 µm) (Dr. A. Maisch, GmbH, Ammerbuch, Germany) was used. The mobile phase for HPLC analysis was 0.1 M citrate-phosphate buffer containing 1.1 M octane sulfonic acid, 0.1 mM ethylenediaminetetraacetic acid (EDTA), and 9% solution of acetonitrile (pH = 3.0). Measurements were made using the electrochemical detector LC-4B (BAS, West Lafayette, IN, USA) on a glass–carbonic electrode (+0.85 V) against the reference electrode Ag/AgCl. The turnover of monoamines was expressed as the ratio of tissue concentrations of the primary acidic metabolite (MHPG, DOPAC, HVA, or 5-HIAA) to the parent amine (NA, DA, or 5-HT). Samples from all animals (vehicle-treated and drug-treated) were processed in parallel on the same day for each brain structure. We used a solution that contained 3,4-dihydroxy benzylamine (DHBA) (Sigma, Ronkonkoma, NY, USA), NA (Bioanalytical Systems, Inc., West Lafayette, IN, USA), 3-methoxy-4-hydroxyphenylglycol (MHPG) (Sigma, USA), DA (Sigma, USA), 3,4-dihydroxyphenylacetic acid (DOPAC) (Sigma, USA), homovanillic acid (HVA) (Sigma, USA), 3-methoxytyramine (3-MT) (Sigma, USA), 5-hydroxytryptamine (5-HT) (Calbiochem, San Diego, CA, USA), and 5-hydroxy indole acetic acid (5-HIAA) (Fluka, Radnor, PA, USA) as a standard at a concentration of 500 pmol/mL. The “internal standard” method was used to determine the levels of monoamines and their metabolites in brain structures as the ratio of the peak areas in the standard mixture and sample (nmol/g wet tissue). Samples were registered using the hardware–software complex Multichrom 1.5 (Ampersand, Moscow, Russia). All the reagents used for the analysis were of high purity.

### 4.4. Statistical Analysis

Data were analyzed using the program “STATISTICA Release 7”. A two-way analysis of variance (ANOVA) with repeated measures and one-way ANOVA with the Newman—Keuls post hoc test or a non-parametric equivalent, the Kruskal—Wallis H test (one-way ANOVA by ranks), and the Mann—Whitney U test with Bonferroni corrections were used when appropriate.

## 5. Conclusions

The results of the present study provide the first evidence that L-MET treatment causes anxiolytic and antidepressant effects, which are associated with absence seizure suppression. The correction of the pathologic phenotype in WAG/Rij rats was accompanied by an increase in the content of monoamines and their metabolites, the insufficiency of which, as previously shown, is associated with pathology in this model.

Behavioral, EEG, and biochemical studies indicate the therapeutic efficacy profile of L-MET in the WAG/Rij rat model. The antidepressant-like effect of L-MET is closer to that of the tricyclic antidepressant imipramine than that of the selective serotonin reuptake inhibitor fluoxetine. L-MET appears to be an antidepressant with anxiolytic properties, similar to imipramine. However, in contrast to imipramine, L-MET does not aggravate epileptic seizures in WAG/Rij rats. It is assumed that neurochemical changes in brain structures caused by L-MET underlie its therapeutic effect. However, the exact mechanisms by which L-MET exert its favorable complex effects on anxiety, depression, and associated absence epilepsy await full elucidation.

L-MET is the immediate precursor of SAMe, which is a major donor of methyl groups not only for the synthesis of monoamines in the brain but also for DNA methylation and epigenetic modifications of gene expression. Emerging evidence suggests an important role of epigenetic mechanisms in the pathogenesis of depression and epilepsy as well as in the mechanisms of antidepressant and anti-epileptic effects of drugs [23,24,27,28,30,77,78,79,80,81]. Epigenetic mechanisms also contribute to the development of absence epilepsy, comorbid depression, and anti-absence drug medication in the WAG/Rij rat model [13,20,29]. Further in-depth studies are still required to understand the epigenetic mechanisms of L-MET action and to assess the potential utility of L-MET and/or other methyl-group donors as a new therapeutic approach for the treatment of anxiety and depression comorbidities in absence epilepsy.

## Figures and Tables

**Figure 1 ijms-24-12425-f001:**
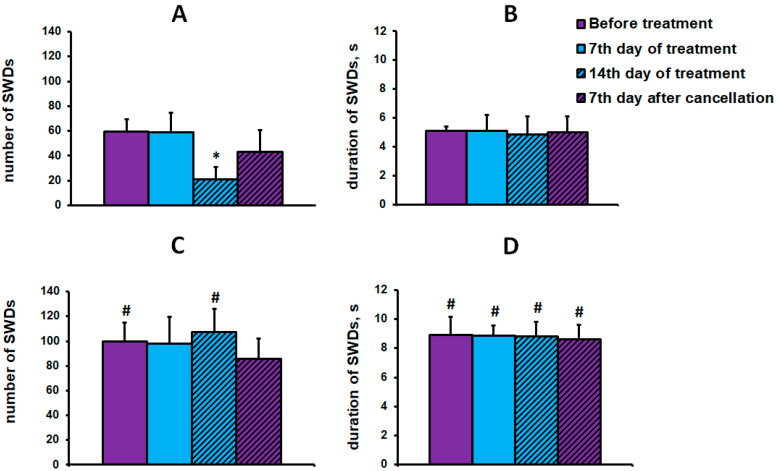
The effect of L-MET on the number and mean duration of SWDs in “sensitive” (**A**,**B**) and “insensitive” (**C**,**D**) subgroups of WAG/Rij rats. (**A**,**C**) The number of SWDs. (**B**,**D**) The mean duration of SWDs. Values are the mean ± standard error of the mean (M ± S.E.M.). * indicates *p* < 0.05 compared with the initial level (before treatment), # indicates *p* < 0.05 in the “insensitive” subgroup compared with the corresponding values in the “sensitive” subgroup.

**Figure 2 ijms-24-12425-f002:**
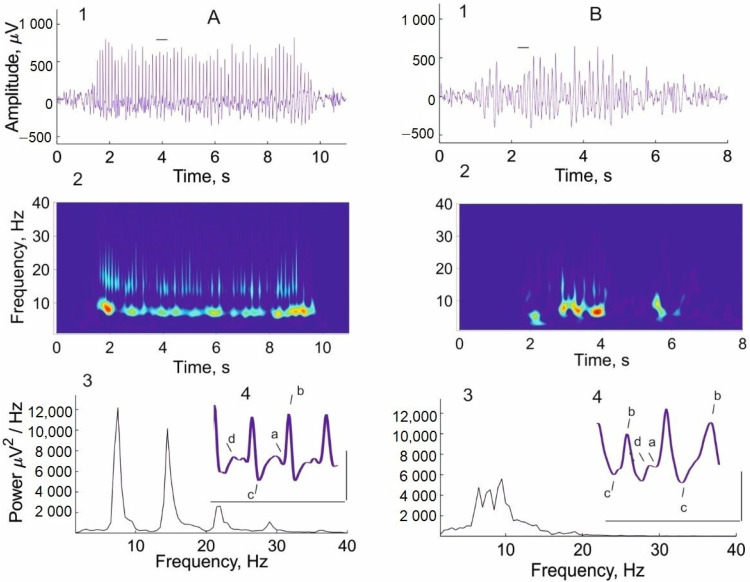
A typical (mature) SWD recorded in a WAG/Rij rat before L-MET treatment (**A**) and an immature discharge recorded on the 14th day of L-MET administration (**B**). 1—examples of epileptic discharges (abscissa—time, s; ordinate—amplitude of SWD, μV); 2—wavelet spectrograms of SWDs reflecting their time–frequency dynamics (abscissa—time, s; ordinate—frequency, Hz); 3—spectral power density of SWDs evaluated by the Welch method using Fast Fourier Transform (abscissa—frequency, Hz; ordinate—spectral power density, μV2/Hz); 4—an expanded presentation of SWD fragments marked by a horizontal line above the discharges to illustrate their waveform morphology: a—positive transient (PT) early, b—spike 2, c—PT late, d—wave. Scale bars indicate a time of 1 s (abscissa) and an amplitude of 500 μV (ordinate).

**Figure 3 ijms-24-12425-f003:**
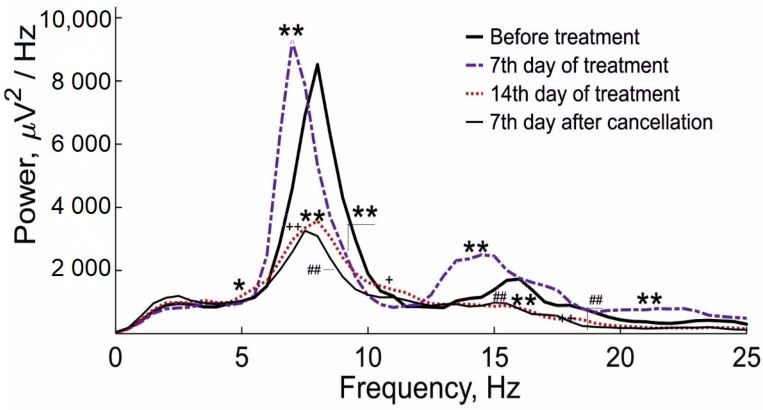
Averaged spectrograms of SWDs in a sensitive subgroup of WAG/Rij rats before treatment, on the 7th day of treatment, on the 14th day of treatment, and on the 7th day after L-MET cancellation, calculated by the Welch method using the Fast Fourier Transform procedure. The Mann–Whitney test revealed significant differences in the spectrograms on the 14th day of treatment and on the 7th day after L-MET cancellation compared with the initial level (≥25 discharges per stage of the experiments in n = 6 rats). * indicates *p* < 0.05 and ** indicates *p* < 0.01 at the frequencies of 6.5–7, 8–10, 13–15, and 20–23 Hz on the 7th day of L-MET treatment compared with before treatment; ++ indicates *p* < 0.01 at the frequencies of 7.5–10.5 and 15–25 Hz; + indicates *p* < 0.05 at the frequency of 5.5 Hz on the 14th day of L-MET treatment compared with before treatment; ## indicates *p* < 0.01 at the frequencies of 6.5–10, 15.5–16.5, and 17.5–21 on the 7th day of L-MET cancellation compared with before treatment. *p* < 0.05—without Bonferroni correction; *p* < 0.01—with Bonferroni correction (312), which is the multiplication of the number of analyzed frequencies (52) in the spectrum with a frequency resolution of 0.5 Hz by the number of comparisons (6).

**Figure 4 ijms-24-12425-f004:**
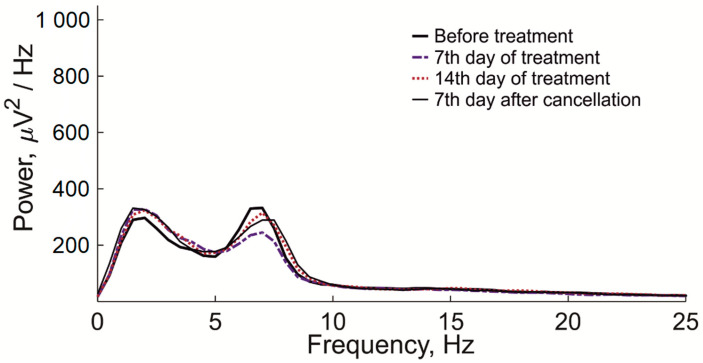
Averaged spectrograms of the background (interictal) EEG in WAG/Rij rats before L-MET treatment, on the 7th day of treatment, on the 14th day of treatment, and on the 7th day after L-MET cancellation, calculated by the Welch method using the Fast Fourier Transform procedure. The Mann–Whitney test revealed no significant effects of L-MET on the averaged spectral power of the background EEG at any of the frequencies (0–25 Hz) analyzed in WAG/Rij rats.

**Figure 5 ijms-24-12425-f005:**
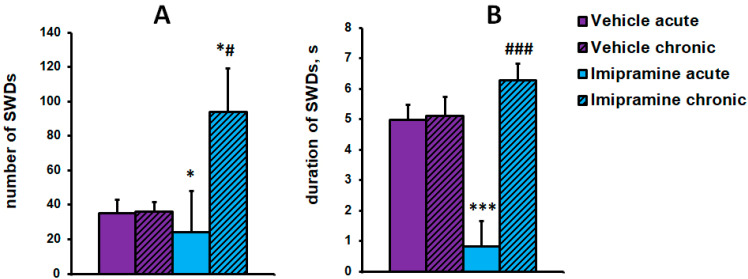
The effect of acute and chronic administration of the antidepressant imipramine and vehicle on the number and mean duration of SWDs in WAG/Rij rats. (**A**) The number of SWDs. (**B**) The mean duration of SWDs. * indicates *p* < 0.05 and *** indicates *p* < 0.001 compared with acute and chronic vehicle treatment; # indicates *p* < 0.05 and ### indicates *p* < 0.001 compared with acute imipramine administration.

**Figure 6 ijms-24-12425-f006:**
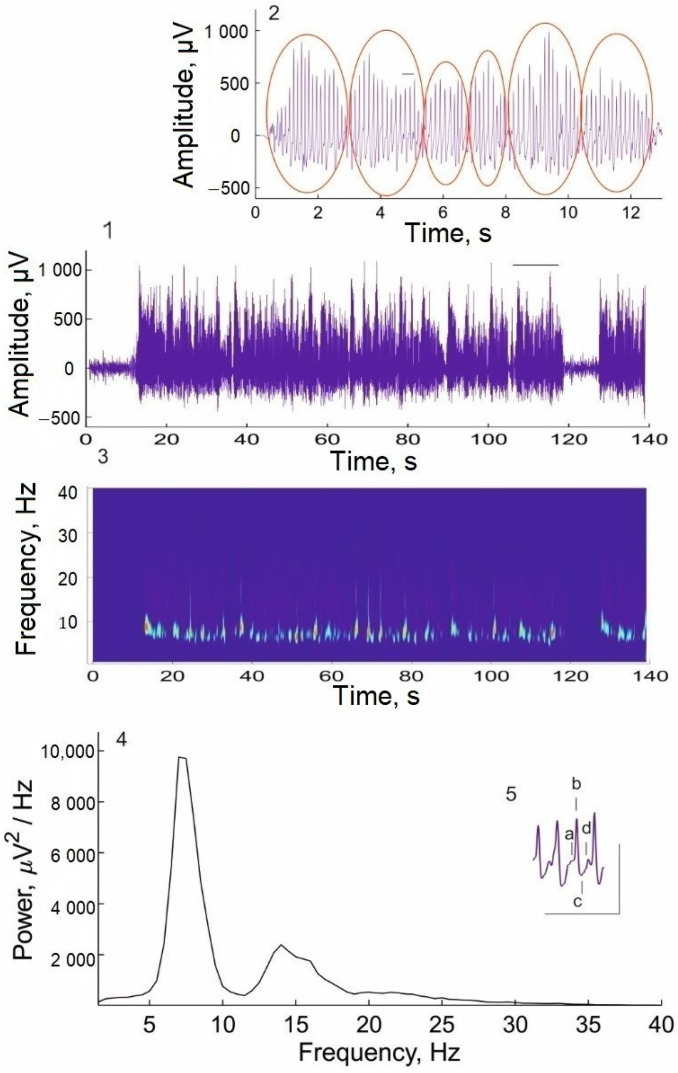
An example of a long SWD recorded in a WAG/Rij rat after chronic (14-day) administration of imipramine at a dose of 15 mg/kg. 1—the long (about 110 s) SWD on an EEG recording; 2—an expanded presentation of the SWD fragment marked by a horizontal line above the SWD (1) to demonstrate the fusion of individual SWDs (marked with red ellipses) into one long SWD; 3—wavelet spectrogram of the long SWD; 4—the spectral power density of the long SWD; 5—an expanded presentation of the SWD fragment marked by a horizontal line above the SWD (2) to illustrate its waveform morphology: a—positive transient (PT) early; b—spike 2, c—PT late, d—wave. Scale bars indicate a time of 500 ms (abscissa) and an amplitude of 1000 µV (ordinate).

**Figure 7 ijms-24-12425-f007:**
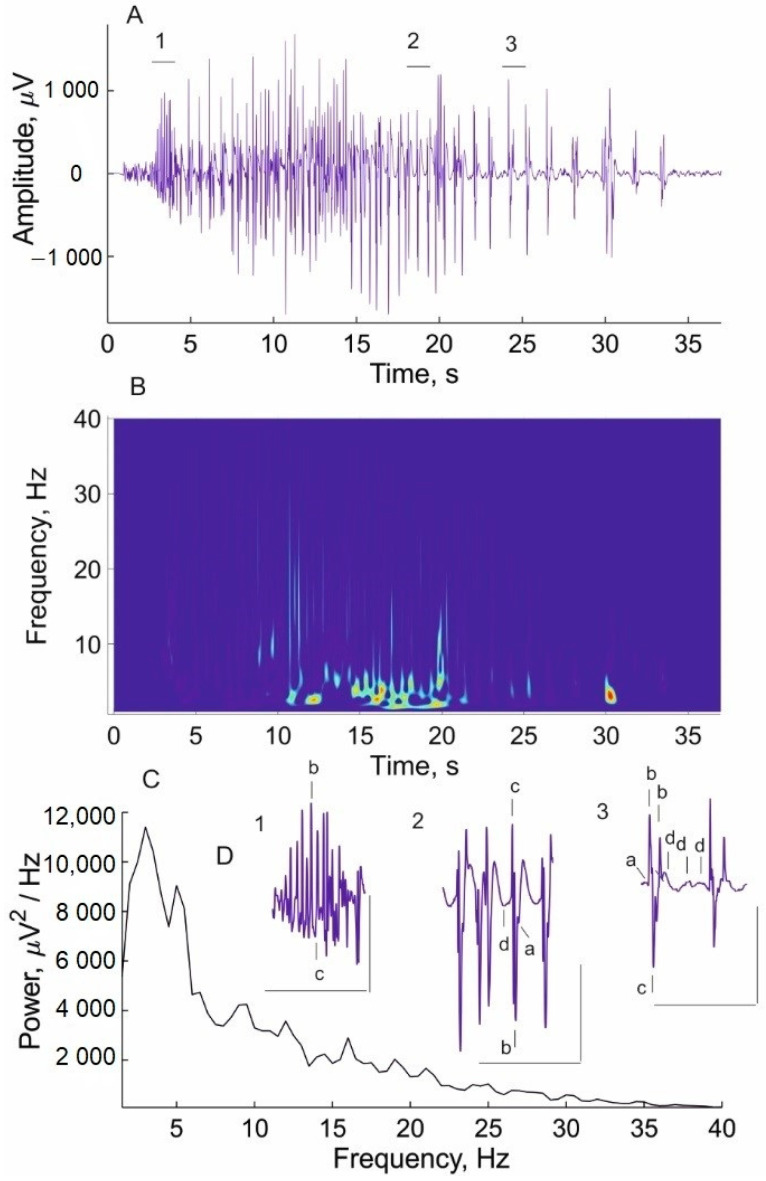
An example of an atypical (abnormal) epileptic discharge caused in a WAG/Rij rat by a two-fold higher dose of imipramine (30 mg/kg) administration for 4 days. (**A**) An abnormal discharge. (**B**) Its wavelet spectrogram. (**C**) The spectral power density of the abnormal discharge. (**D**) An expanded presentation of the morphological characteristics of 3 consecutive parts (1, 2, 3) of the abnormal epileptic discharge (marked by a horizontal line in A) to illustrate the heterogeneity of the discharge morphology: a—PT early, b—spike, c—PT late, d—wave. Scale bars indicate a time of 2.5 s (abscissa) and an amplitude of 1000 µV (ordinate).

**Figure 8 ijms-24-12425-f008:**
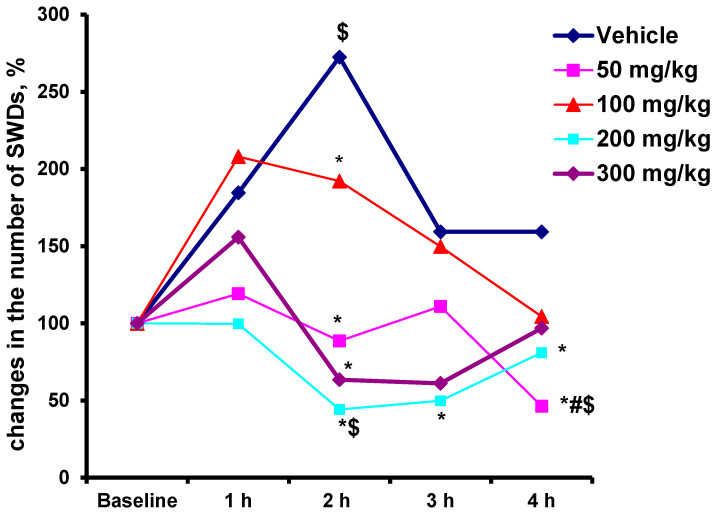
Percentage changes in the number of absence seizures in WAG/Rij rats after intraperitoneal single injection of different doses of L-MET and saline compared with the baseline level taken as 100%. Abscissa—the time after administration of drugs (hours); ordinate—the number of SWDs as a percentage of the initial level. Each value represents the mean value in three rats. * indicates *p* < 0.05 compared with vehicle; # indicates *p* = 0.05 compared with L-MET at a dose of 100 mg/kg; $ indicates *p* < 0.05 compared with the baseline level.

**Table 1 ijms-24-12425-t001:** The effect of L-MET on anxiety measures in WAG/Rij rats.

Behavioral Measures	WAG/Rij Vehicle (n = 10)	WAG/Rij L-MET (n = 6)
**Open-field test**		
Latency to leave the center, s	4.7 ± 0.7	2.5 ± 0.3 *
Number of squares crossed	61.5 ± 4.9	86.3 ± 8.7 *
Number of assisted rearings	3.3 ± 0.7	5.2 ± 1.33
Number of unassisted rearings	2.1 ± 0.4	5.3 ± 1.5 *
Number of center entries	0.9 ± 0.5	4.3 ± 0.8 **
Number of groomings	3.3 ± 0.9	4.2 ± 1.2
Number of boli	1.4 ± 0.4	0.3 ± 0.3 +
**Light–dark choice test**		
Time in the light, s	18.1 ± 4.4	43.3 ± 14.1 *
Number of risk assessments	5.3 ± 0.6	2.7 ± 1.0 *
Number of transitions	2.2 ± 0.4	4.0 ± 0.9 +
**Elevated plus-maze test**		
Time in open arms, s	28.6 ± 5.3	69.7 ± 21.9 *
Number of transitions between arms	4.7 ± 0.5	4.1 ± 0.8
Number of risk assessments	3.6 ± 0.6	1.8 ± 0.6
Number of hangings from the open arms	2.3 ± 0.7	7.5 ± 2.1 *
Number of rearings in the closed arms	3.6 ± 0.6	1.8 ± 0.7 +
Number of boli	2.2 ± 0.8	0.8 ± 0.5 *

Values are the mean ± standard error of the mean (M ± S.E.M.). * indicates *p* < 0.05; ** indicates *p* < 0.01; + indicates 0.05 < *p* < 0.1 (tendency) compared with a vehicle-treated control group.

**Table 2 ijms-24-12425-t002:** The effects of 14-day administration of L-MET, imipramine, and fluoxetine on the behaviour of WAG/Rij rats in the forced swimming test.

Behavioral measures	Treatment option	L-MET 50 mg/kg (n = 15)	Imipramine 15 mg/kg (n = 16)	Fluoxetine 15 mg/kg (n = 14)
Immobility time, s	Vehicle Drug	209.6 ± 5.4 176.2 ± 17.5 * (84.1%)	210.5 ± 5.8 176.3 ± 6.9 *** (83.8%)	213.0 ± 7.5 180.9 ± 7.4 ** (84.9%)
The first episode of active swimming, s	Vehicle Drug	34.3 ± 3.8 48.3 ± 3.6 * (140.8%)	28.5 ± 2.6 50.1 ± 2.2 *** (175.8%)	33.9 ± 2.7 36.3 ± 2.0 (107.1%)
Swimming, s	Vehicle Drug	56.1 ± 6.8 75.5 ± 14.9 (134.6%)	61.0 ± 5.9 73.6 ± 6.4 (120.7%)	53.1 ± 8.4 82.9 ± 6.7 * (156.1%)
Number of dives	Vehicle Drug	1.1 ± 0.4 4.0 ± 0.7 ** (363.6%)	0.9 ± 0.4 3.3 ± 0.3 ** (363.7%)	2.0 ± 0.5 2.3 ± 0.4 (115%)

Values are the mean ± standard error of the mean (M ± S.E.M.). * *p* < 0.05, ** *p* < 0.01 and *** *p* < 0.001 compared with vehicle-treated control group. Numbers in parenthesis indicate the values in % relative to the control (vehicle-treated) group, which is taken as 100%.

**Table 3 ijms-24-12425-t003:** The effects of 14-day administration of L-MET and the antidepressant imipramine on the amplitude and asymmetry index of SWDs compared with the effect of saline.

Drug	Amplitude, µV	Asymmetry Index, %
L-MET, 50 mg/kg	666.1 ± 36.1 ***^###^	61.4 ± 1.9 ***^###^
Imipramine, 15 mg/kg	814.1 ± 10.2 ^##^	69.1 ± 0.3
Vehicle	889.7 ± 13.8	69.4 ± 0.5

Values are the mean ± standard error of the mean (M ± S.E.M). *** *p* < 0.001 compared with imipramine-treated group; ^##^
*p* < 0.01 and ^###^
*p* < 0.001 compared with a vehicle-treated group.

**Table 4 ijms-24-12425-t004:** The effect of L-MET on monoamines and their metabolite content in brain structures of WAG/Rij rats.

Biochemical Measures, nmol/g Tissue	Brain Structures				
	Prefrontal Cortex	Nucleus Accumbens	Striatum	Hypothalamus	Hippocampus
NA	1.31 ± 0.07	2.10 ± 0.42	0.71 ± 0.07	5.11 ± 0.17	1.44 ± 0.09
	**1.61 ± 0.07 ***	**1.65 ± 0.29**	**0.73 ± 0.11**	**6.20 ± 0.21 ****	**1.43 ± 0.08**
MHPG	ND	0.03 ± 0,01	0.05 ± 0.02	0.39 ± 0.02	ND
		**0.09 ± 0.03 +**	**0.07 ± 0.02**	**0.58 ± 0.03 *****	
DA	0.76 ± 0.36	10.63 ± 1.80	22.68 ± 0.80	0.80 ± 0.04	0.18 ± 0.05
	**0.53 ± 0.30**	**14.50 ± 1.98 +**	**26.03 ± 1.45 +**	**1.00 ± 0.08+**	**0.80 ± 0.36 +**
DOPAC	0.09 ± 0.03	1.34 ± 0.21	3.06 ± 0.07	0.13 ± 0.01	0.04 ± 0.01
	**0.07 ± 0.02**	**2.02 ± 0.23 ***	**3.54 ± 0.20 +**	**0.16 ± 0.02**	**0.12 ± 0.04 +**
HVA	0.13 ± 0.03	0.80 ± 0.09	1.21 ± 0.06	0.14 ± 0.03	0.03 ± 0.01
	**0.13 ± 0.03**	**1.00 ± 0.20**	**1.60 ± 0.05 *****	**0.12 ± 0.02**	**0.08 ± 0.02**
3-MT	0.06 ± 0.02	0.33 ± 0.07	0.60 ± 0.03	0.02 ± 0.00	0.03 ± 0.01
	**0.08 ± 0.02**	**0.50 ±0.09**	**0.72 ± 0.08**	**0.03 ± 0.00**	**0.05 ± 0.01 +**
DOPAC/DA	0.18 ± 0.03	0.13 ± 0.00	0.14 ± 0.00	0.16 ± 0.00	0.21 ± 0.03
	**0.22 ± 0.03**	**0.15 ± 0.02**	**0.14 ± 0.00**	**0.16 ± 0.01**	**0.23 ± 0.03**
HVA/DA	0.35 ± 0.14	0.08 ± 0.01	0.05 ± 0.00	0.17 ± 0.03	0.22 ± 0.04
	**0.64 ± 0.23**	**0.07 ± 0.01**	**0.06 ± 0.00 ***	**0.11 ± 0.02**	**0.21 ± 0.06**
5-HT	1.37 ± 0.10	2.53 ± 0.37	1.78 ± 0.06	3.16 ± 0.13	1.26 ± 0.05
	**1.34 ± 0.08**	**1.71 ± 0.34**	**1.81 ± 0.07**	**3.12 ± 0.14**	**1.23 ± 0.03**
5-HIAA	1.02 ± 0.10	1.74 ± 0.26	1.88 ± 0.15	2.10 ± 0.12	1.34 ± 0.13
	**1.01 ± 0.08**	**1.63 ± 0.23**	**2.10 ± 0.14**	**2.25 ± 0.11**	**1.43 ± 0.07**
5-HIAA/5-HT	0.77 ± 0.09	0.70 ± 0.04	1.07 ± 0.10	0.67 ± 0.03	1.06 ± 0.09
	**0.75 ± 0.04**	**1.30 ± 0.35 +**	**1.16 ± 0.06**	**0.72 ± 0.02 +**	**1.16 ± 0.05**

Values are the mean ± standard error of the mean (M ± S.E.M.) for vehicle-treated (upper line, n = 8) and L-MET-treated (bottom line, highlighted in bold, n = 8) mice. NA—noradrenaline; MHPG—3-methoxy-4-hydroxyphenylglycol; DA—dopamine; DOPAC—3,4-dihydroxyphenylacetic acid; HVA—homovanillic acid; 3-MT—3-methoxytyramine; 5-HT—serotonin; 5-HIAA—5-hydroxy indole acetic acid. * indicates *p* < 0.05; ** indicates *p* < 0.01; *** indicates *p* < 0.001; + indicates 0.05 < *p* ≤ 0.1 (tendency). ND—not defined. Differences are highlighted by darkening.

## Data Availability

The experimental data are fully presented in the manuscript. Additional information may be provided upon reasonable request.

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
