# Peer review of "Antidepressant and Anxiolytic Effects of L-Methionine in the WAG/Rij Rat Model of Depression Comorbid with Absence Epilepsy"

_ijms, 2023, doi:10.3390/ijms241512425_

Round 1

Reviewer 1 Report

Review - manuscript ijms-2503237

Depression is the most frequent comorbidity of different forms of epilepsy, including absence epilepsy. The search for new effective drugs is currently underway.

SAMe, the universal donor of methyl groups for DNA methylation and brain monoamine synthesis, is known to have high antidepressant activity. The authors of the ijms-2503237 manuscript set themselves the goal of their research to check whether L-methionine, a precursor of SAM, has a similar effect. They conducted experimental research on the WAG/Rij rat model of depression comorbid with absence epilepsy.

L-MET was administered intraperitoneally to rats at a dose of 50 mg/kg for 14 days. This dose was chosen on the basis of previous studies in which the authors determined that it raises SAMe levels in the brain. In addition to chronic administration of L-MET, rats were also injected with single doses of L-MET (50, 100, 200 and 300 mg/kg b.w.; i.p.).

Imipramine hydrochloride (tricyclic antidepressant) and Fluoxetine (selective serotonin reuptake inhibitor) were given as reference drugs (15 mg/kg, i.p., 14 days).

As part of the verification of L-MET activity, the authors performed a number of behavioural tests that allowed to assess the level of anxiety and depressive-like symptoms (light-dark choice test, open field test, elevated plus-maze test, forced swimming test) as well as EEG registration. Post-mortem neurochemical studies were also performed - i.e. concentrations of catecholamines (DA, NA), serotonin and their metabolites (DOPAC, HVA, MHPG, 5-HIAA) were measured in the brain.

Based on the obtained results, the authors concluded that L-MET reduces the level of anxiety and depression in WAG/Rij rats and suppresses accompanying epileptic seizures. This effect was associated with an increase in the levels of monoamines (DA and NA) and their metabolites (DOPAC, HVA and MHPG) in several brain structures. These are only the first studies, so it is too early to conclude that the above-mentioned neurochemical changes are the molecular mechanism of action of L-MET.

In my opinion, after L-MET injections, it would be useful to examine the activity of enzymes metabolizing DA, NA and 5-HT (i.e. MAO A and B, COMT) in the rat brain.

To sum up, the conducted studies were well designed and comprehensively performed - I have no objections to that, apart from the above-mentioned comment.

However, the manuscript requires editorial corrections. I also have reservations about the fact that Dr. Karine Sarkisova (corresponding author) is a co-author of 15 cited articles (including the first author of 13 articles). I understand that the topic of the manuscript is a continuation of Dr. Sarkisowa's research, but quoting so many of the author's own articles seems to me to be an exaggeration.

Detailed notes:

1.     In my opinion, the number of self-citations should be reduced.

2.     Line 146 – the title of  Table 1 should be moved to the next page.

3.     Table 1 requires appropriate formatting, the headings of the second and third columns should be placed directly above the data presented in them.

4.     Tables 2 and 3 are completely illegible - they also need to be formatted.

5.     Figure 1 has a very bad graphic quality, some of its elements have been ungrouped (markings of individual graphs: letters A, B, C, D; statistical significance marks).

6.     Figures 2, 3, 4, 6 and 7 - graphical unification would be useful, e.g. please use the same font size.

7.     Figure 5 – please increase the font size. Figure 5 should look the same as Figure 1.

8.     The record of the literature requires scrupulous verification. Writing some DOI access paths jumped to the next line unnecessarily (e.g. record 6, 21, 27).

9.     The section where abbreviations are explained should also be corrected (lines 811-820).

Author Response

  1. Based on the obtained results, the authors concluded that L-MET reduces the level of anxiety and depression in WAG/Rij rats and suppresses accompanying epileptic seizures. This effect was associated with an increase in the levels of monoamines (DA and NA) and their metabolites (DOPAC, HVA and MHPG) in several brain structures. These are only the first studies, so it is too early to conclude that the above-mentioned neurochemical changes are the molecular mechanism of action of L-MET.

Yes, we agree that these are only the first studies and therefore it is too early to conclude that the identified neurochemical changes are the molecular mechanism of L-MET action. This is just an assumption that requires further research. Biochemical changes caused by L-MET only indicate the possible involvement of molecular mechanisms underlying these changes, primarily alterations in the expression of genes related to monoamine metabolism. We have replaced the term “molecular mechanism” with the term “neurochemical mechanism” in accordance with your remark.

  1.  In my opinion, after L-MET injections, it would be useful to examine the activity of enzymes metabolizing DA, NA and 5-HT (i.e. MAO A and B, COMT) in the rat brain.

Yes, it would be really useful in the future to examine the activity of enzymes metabolizing DA, NA and 5-HT (i.e. MAO A and B, COMT) in the WAG/Rij rat brain. The increase in the level of DA, NA and their metabolites caused by L-MET in almost all studied brain structures indicates the effect of L-MET on tyrosine hydroxylase and it would be necessary first of all to examine the expression of the tyrosine hydroxylase, the rate-limiting enzyme in the synthesis of monoamines, along with MAOA, MAOB and COMT. However, if L-MET affected MAOA, MAOB and COMT, similar to conventional antidepressants, then the content of DA and NA metabolites (DOPAC, HVA, MHPG) should have decreased, but in our study the content of these metabolites increased. This probably means that the mechanism of action of L-MET differs from that of classical antidepressant drugs, which commonly inhibit MAOA, MAOB and COMT. A curious coincidence: the SAME treatment, similar to L-MET, produced an increase in the content of DA, DOPAC, HVA and an improvement in depressive-like behavior. At the same time, the expression of MAOA, MAOB and COMT genes decreased, not increased, as the authors expected (Becker et al., 2022).

 To sum up, the conducted studies were well designed and comprehensively performed - I have no objections to that, apart from the above-mentioned comment.   

However, the manuscript requires editorial corrections. I also have reservations about the fact that Dr. Karine Sarkisova (corresponding author) is a co-author of 15 cited articles (including the first author of 13 articles). I understand that the topic of the manuscript is a continuation of Dr. Sarkisowa's research, but quoting so many of the author's own articles seems to me to be an exaggeration.

Detailed notes:

  1. In my opinion, the number of self-citations should be reduced.

The number of self-citation has been reduced: 3 self-citations have been removed from the list of references. We could not remove more, as we would have to delete the text where the links to the data of these articles are given.

  1. Line 146 – the title of Table 1 should be moved to the next page.

It has been corrected.

  1. Table 1 requires appropriate formatting, the headings of the second and third columns should be placed directly above the data presented in them.

Table 1 has been formatted. The second and third columns have been placed directly above the data presented in them.

  1. Tables 2 and 3 are completely illegible - they also need to be formatted.

            Tables 2 and 3 have been formatted.

  1. Figure 1 has a very bad graphic quality, some of its elements have been ungrouped (markings of individual graphs: letters A, B, C, D; statistical significance marks).

Figure 1 has been improved.

  1. Figures 2, 3, 4, 6 and 7 - graphical unification would be useful, e.g. please use the same font size.

Graphical unification has been made. In the figures 2, 3, 4 and 7 the same font size was used.

  1. Figure 5 – please increase the font size. Figure 5 should look the same as Figure 1.

The font size in the figure 5 has been increased.

  1. The record of the literature requires scrupulous verification. Writing some DOI access paths jumped to the next line unnecessarily (e.g. record 6, 21, 27).

The list of references has been checked and corrected where necessary.

  1. The section where abbreviations are explained should also be corrected (lines 811-820).

            The section with abbreviations has been corrected.

Reviewer 2 Report

The current study by Sarkisova et al. is a novel, original, well-conducted finding and represents an interesting addition to the literature. However, some issues need to be addressed before acceptance for publication.

The review of Tallarico et al. has well described the role of antidepressants in preclinical models of epilepsy as well as their impact on PWE. By virtue of this, it should be added to the citations.

Authors should improve the quality and layout of figures and tables. For instance, the labels and symbols should be centered. Moreover, the labels of some figures are different from others. Furthermore, the authors should use different colors to indicate the four experimental groups in Figure 5. Please can authors well-check and uniform all figures and tables? 

The authors should also further discuss why the dose of 50 mg has an increased efficacy compared to 100 mg, likewise, why the dose of 200 mg is better than 300 mg. Moreover, the symbols of significance should be inserted in this figure.

Why have the authors only evaluated the changes in the number of SWDs? It would be interesting also evaluate the efficacy of this compound on the duration of SWDs.

Author Response

Dear Reviewer 2, thank you for your comments and remarks that helped us to improve the weak points of our article.

The current study by Sarkisova et al. is a novel, original, well-conducted finding and represents an interesting addition to the literature. However, some issues need to be addressed before acceptance for publication.

1. The review of Tallarico et al. has well described the role of antidepressants in preclinical models of epilepsy as well as their impact on PWE. By virtue of this, it should be added to the citations.

            The review of Tallarico et al. has been added to the citations (ref. 71).

2. Authors should improve the quality and layout of figures and tables. For instance, the labels and symbols should be centered. Moreover, the labels of some figures are different from others. Furthermore, the authors should use different colors to indicate the four experimental groups in Figure 5. Please can authors well-check and uniform all figures and tables? 

The quality and layout of figures and tables have been improved.

3. The authors should also further discuss why the dose of 50 mg has an increased efficacy compared to 100 mg, likewise, why the dose of 200 mg is better than 300 mg. Moreover, the symbols of significance should be inserted in this figure.

A). The symbols of significance have been inserted in Figure 8.

B). It is also discussed why the dose of 50 mg/kg was the most effective:

“One important question is why the smallest dose of L-MET was the most effective in suppressing absence seizures. First of all, apparently, because it is this dose of L-MET that increases the SAMe content in the brain, but not in the blood. These data indicate that L-MET causes an increase in the local synthesis of the SAMe in the brain, rather than an increase in the synthesis of SAMe peripherally, with subsequent transport of SAMe from the periphery to the brain [36]. We agree with the authors’ assumption that a small dose of L-MET increases the level of SAMe in the brain because methionine adenosyltransferase, the enzyme that produces SAMe from methionine, is not normally saturated with methionine [36]. Higher doses of L-MET (above the optimal dose) may cause a decrease in the synthesis of SAMe due to substrate inhibition. It is known that many enzymes are inhibited by their substrates, leading to velocity curves that rise to a maximum and then descend as the substrate concentration increases. Substrate inhibition often has important biological functions. For example, substrate inhibition of tyrosine hydroxylase results in a steady synthesis of DA despite large fluctuations in tyrosine due to meals. It is not excluded that substrate inhibition of  methionine adenosyltransferase also leads to a steady synthesis of SAMe in the brain, regardless of fluctuations in the level of methionine” (p.19).

4. Why have the authors only evaluated the changes in the number of SWDs? It would be interesting also evaluate the efficacy of this compound on the duration of SWDs.

The effects of L-MET were evaluated not only on the number of SWDs, but also on the duration of SWDs (see Figure 1). L-MET reduced the number of SWDs, but did not affect the mean duration of SWDs.

Reviewer 3 Report

The authors in this study mainly investigated antidepressant and/or anxiolytic effects of L-methionine (L-MET), a precursor of SAMe, in the WAG/Rij rat model of depression comorbid with absence epilepsy. They found that L-MET reduces the level of anxiety and depression in WAG/Rij rats and suppresses associated epileptic seizures in contrast to conventional antidepressant imipramine. In addition, they found that the antidepressant effect of L-MET was comparable with imipramine and fluoxetine. However, the antidepressant profile of L-MET was more similar to imipramine than to fluoxetine. Their findings suggest that L-MET could serve as a new promising antidepressant drug with anxiolytic properties for the treatment of depression comorbid with absence epilepsy.

I have several major concerns. First, although increased levels of monoamines and their metabolites in several brain structures were presented in the study, the possible molecular or cellular mechanism underlying the beneficial phenotypic effect of L-MET is less explored in current study. Second, the way and the quality of data presentation should be significantly improved before acceptance for publication. Third, the discussion part needs to be more concise and remain focused.

Author Response

Dear Reviewer 3, thank you for your comments and remarks that helped us to improve the weak points of our article.

The authors in this study mainly investigated antidepressant and/or anxiolytic effects of L-methionine (L-MET), a precursor of SAMe, in the WAG/Rij rat model of depression comorbid with absence epilepsy. They found that L-MET reduces the level of anxiety and depression in WAG/Rij rats and suppresses associated epileptic seizures in contrast to conventional antidepressant imipramine. In addition, they found that the antidepressant effect of L-MET was comparable with imipramine and fluoxetine. However, the antidepressant profile of L-MET was more similar to imipramine than to fluoxetine. Their findings suggest that L-MET could serve as a new promising antidepressant drug with anxiolytic properties for the treatment of depression comorbid with absence epilepsy.

I have several major concerns.

  1. First, although increased levels of monoamines and their metabolites in several brain structures were presented in the study, the possible molecular or cellular mechanism underlying the beneficial phenotypic effect of L-MET is less explored in current study.

We agree with your comment. The possible molecular or cellular mechanism underlying the beneficial phenotypic effect of L-MET is less investigated in the current study and needs to be more deeply explored in the future. Biochemical changes caused by L-MET only indicate the possible involvement of molecular mechanisms underlying these changes, primarily alterations in the expression of genes related to monoamine metabolism. Strictly speaking, neurochemical mechanisms have been investigated in this study, so we have replaced the term “molecular mechanism” with the term “neurochemical mechanism”.

  1. Second, the way and the quality of data presentation should be significantly improved before acceptance for publication. Third, the discussion part needs to be more concise and remain focused.

The quality of data presentation was significantly improved.

Round 2

Reviewer 1 Report

The authors revised the manuscript addressing all my comments.

Reviewer 2 Report

From my perspective, this manuscript's revised version can be accepted for publication.

Reviewer 3 Report

The manuscript was improved after revision. I have no more major concerns on its publication.